# Advances in the Pathophysiology of Thrombosis in Antiphospholipid Syndrome: Molecular Mechanisms and Signaling through Lipid Rafts

**DOI:** 10.3390/jcm12030891

**Published:** 2023-01-23

**Authors:** Antonella Capozzi, Valeria Manganelli, Gloria Riitano, Daniela Caissutti, Agostina Longo, Tina Garofalo, Maurizio Sorice, Roberta Misasi

**Affiliations:** Department of Experimental Medicine, “Sapienza” University of Rome, 00161 Rome, Italy

**Keywords:** APS, β2-GPI, aPLs, lipid rafts, TLR4 pathways, therapeutic targets

## Abstract

The pathological features of antiphospholipid syndrome (APS) are related to the activity of circulating antiphospholipid antibodies (aPLs) associated with vascular thrombosis and obstetric complications. Indeed, aPLs are not only disease markers, but also play a determining pathogenetic role in APS and exert their effects through the activation of cells and coagulation factors and inflammatory mediators for the materialization of the thromboinflammatory pathogenetic mechanism. Cellular activation in APS necessarily involves the interaction of aPLs with target receptors on the cell membrane, capable of triggering the signal transduction pathway(s). This interaction occurs at specific microdomains of the cell plasma membrane called lipid rafts. In this review, we focus on the key role of lipid rafts as signaling platforms in the pathogenesis of APS, and propose this pathogenetic step as a strategic target of new therapies in order to improve classical anti-thrombotic approaches with “new” immunomodulatory drugs.

## 1. Introduction

Antiphospholipid syndrome (APS) is a multisystemic disorder first described in the 1980s as an autoantibody-induced thrombophilia. The pathological features of APS are related to the activity of circulating antiphospholipid antibodies (aPLs) associated with vascular thrombosis and obstetric complications [1,2,3]. APS usually occurs in the fourth decade of life, and patients can be classified into two groups: primary disease, if it presents in the absence of another autoimmune disorders, or secondary disease, if it is associated with a concomitant connective disease, most commonly systemic lupus erythematosus (SLE) [4]. In patients with APS, thrombosis can affect arterial, venous and microvascular circuits; however, arterial ones are more severe and more commonly found in male than female patients. They frequently occur in the cerebral circulation, causing stroke and transient ischemic attack [5,6].

Recurrent miscarriage is the main obstetric complication of obstetrical APS (OAPS); however, other events, such as unexplained fetal death or premature birth related to placental insufficiency, eclampsia or preeclampsia, may also represent adverse events associated with OAPS [5,6,7].

Beyond vascular events and pregnancy morbidities, the clinical spectrum of the APS may include additional extra-criteria manifestations that cannot be explained solely by the onset of a prothrombotic state. The first description of APS by Hughes and colleagues already included the involvement of the nervous system [8]. The pathogenesis concerns not only thrombo-occlusive events, but refers to neurological manifestations, such as migraine, epilepsy, chorea, and cognitive dysfunctions, probably as a consequence of the direct interaction of aPLs with nervous tissues [9,10,11]. The presence of aPLs is often associated with hematological manifestations, particularly in patients with idiopathic thrombocytopenic purpura (ITP), where aPLs may be responsible for the induction of thrombocytopenia [12,13]. Among the cutaneous manifestations of APS, the most common is livedo reticularis, but pseudovasculitic lesions, necrotic skin ulcers and digital gangrene can also be found [5,14,15]. A minority of patients (less than 1%) may develop a severe variant of the disease, defined catastrophic APS (CAPS) and described as an acute multiorgan dysfunction, with a rapid onset of microvascular thrombosis in at least three organs; a mortality rate of more than 50% is reported in cases not promptly treated [16,17].

The antigens recognized by aPLs are phospholipids, phospholipid/protein complexes and phospholipid-binding proteins, among which β2-glycoprotein I (β2-GPI) is now recognized as the major autoantigen with the highest antibody-binding activity in APS patients [18,19,20,21].

The association between aPLs and thrombosis is now well documented, just as it is ascertained that these autoantibodies are not only markers of disease but are even responsible for the induction of a procoagulant phenotype and therefore for triggering the pathophysiology of APS. However, although several mechanisms have been proposed to describe the pathogenesis of APS, it is difficult task, due to the heterogeneity of autoantibody profiles and diversity in the potential effector functions of aPLs [22,23,24].

In this review, we describe the pathogenic mechanisms of APS through the key role of lipid rafts as signaling platforms, suggesting this pathogenetic step as a strategic target of new therapies.

## 2. Mechanisms in the Pathophysiology of APS

Evidence of the thrombogenic activity of aPLs has been provided by both in vitro and in vivo studies; however, despite the presence of aPLs in the circulation, thrombotic manifestation is not always observed in the patients [25,26]. Moreover, several results in experimental animal models demonstrate that the infusion of aPLs alone does not cause spontaneous thrombotic complications; on the contrary, the appearance of a thromboinflammatory state can be observed after small injuries to the vessel wall or infusion of lipopolysaccharides and other immunostimulants which lead to a disruption of the vascular system. These considerations suggest the “two-hit hypothesis” concept to better clarify the pathogenetic role of aPLs in the mechanisms underlying the development of thrombosis in APS patients. According to the “two-hit hypothesis”, the aPLs provide the first hit, inducing a thrombophilic state, but a second condition (second hit) such as trauma, surgery, infection, or other inflammatory stimulus is required for thrombosis to take place [27,28,29,30]. Thrombus formation is the most studied aspect of APS pathophysiology, and the procoagulant phenotype observed in the disease has been described as a result of the synergistic activation of many elements, where inflammation represents a central pathogenetic factor that acts as a mediator between the coagulation dysfunction and thrombotic manifestations. Several studies have shown that aPLs exert their effects through activation of various cells (endothelial cells, monocytes, platelets, endometrial and decidual cells) [31], as well as a complement system, coagulation factors and inflammatory mediators [32]. These pathogenetic mechanisms underlying clinical manifestations of APS can be exploited as targets of immunomodulatory therapeutic strategies to be used in APS patients, as summarized in Figure 1.

aPLs activate endothelial cells to release several proinflammatory cytokines and to express adhesion molecules, involving toll-like receptors (TLRs) and LDL receptor related protein 8 (LRP8) [33,34,35]. Moreover, anti-β2-GPI antibodies may activate monocytes via several signaling pathway kinases and through engagement of various TLRs, leading to a proinflammatory phenotype [36,37]. Pathogenic aPLs also induce platelet activation and aggregation, where TLRs, LRP8 and the GPIbα subunit of the GPIb-V-IX play a role as candidate receptors responsible for signaling pathway activation [38,39].

A central player in the development of antibody-mediated thrombosis in APS is the tissue factor (TF), a primary cellular initiator of the extrinsic coagulation cascade. An up-regulation of TF in endothelial cells, monocytes and platelets following anti-β2-GPI antibodies treatment has been demonstrated [40,41,42]. Furthermore, the results of our paper showed that the expression of TF was significantly increased in platelets of patients with APS compared to those of healthy donors [43].

aPLs may induce the formation of neutrophil extracellular traps (NETs), which are mostly known to trap pathogens, but they also play a prothrombotic role activating platelets and clotting factors. It was highlighted that monoclonal anti-β2-GPI antibodies were able to promote NETs release in vitro [44,45,46]. In further APS case studies, increased release of NETs in the neutrophils was demonstrated, and serum samples have high levels of circulating NETs as well as being defective in degrading NETs. Recently, anti-NET antibodies have been found in sera from patients with primary APS [47,48].

In addition, the thrombogenic effect of aPLs may involve the activation of the complement system, through both the alternative and the classical pathway. Experiments in mice showed that C3 and C5 activation was necessary for aPL-induced thrombosis [49,50]. There is evidence that C5a factor binds to endothelial cells, resulting in increased neutrophil adhesion, TF expression, and release of other procoagulant molecules [51,52,53].

Many studies have described the direct activity of aPLs in association with pregnancy complications as part of the pathogenesis of OAPS. In particular, aPLs may interact with trophoblast cells and consequently apoptosis, abnormal cell differentiation and decreased secretion of human chorionic gonadotropin may be observed. Inflammation plays a central role in inducing pregnancy morbidity in OAPS patients; in fact, at the utero-placental interface, aPLs induce a proinflammatory phenotype in the vasculature. These molecular mechanisms lead to endothelial injury and activation of monocytes and neutrophils involving the activation of the complement system [54,55,56,57].

The heterogeneity of the mechanisms underlying APS may be explained by the different aPL profiles, which also include extra-antibodies beyond those routinely tested [58,59]; thus, the clinical characteristics of the disease in an individual are due to different autoantibody specificities, together with genetic comorbidities and acquired risk factors.

At the end, the heterogeneity of the mechanisms underlying APS may be related to the different signal transduction pathways triggered by aPLs. Recently, increasing evidence suggests that they are driven by specific microdomains on the cell plasma membrane termed lipid rafts.

## 3. Lipid Rafts in the Immune Signaling

Cellular membranes are not homogenous mixtures of lipids and proteins, but some, such as free cholesterol and glycosphingolipids segregate into lipid rafts [60,61]. Since several investigations found that glycosphingolipid clusters are resistant to detergent extraction, these specialized subdomains have also designed detergent-resistant membranes (DRMs). In any case, lipid rafts are now universally defined as small (10–200 nm) heterogeneous membrane domains enriched in glycosphingolipid and cholesterol that regulate cellular polarity and vesicular traffic as well as cell signaling pathways [61,62].

These microdomains are well known for their role in receptor signaling on the plasma membrane and are essential to cellular functions such as signal transduction and spatial organization of the cellular membrane. As a result, the structural properties of microdomains can promote compartmentalization of plasma membrane proteins that have a higher affinity for the liquid-ordered phase, while they will exclude proteins that have higher affinity for the liquid-disordered phase. In this scenario, some protein–protein interactions will be promoted, while others will be prevented. A complex network of lipid–protein, lipid–lipid, and protein–protein interactions contribute to the activation of a variety of signal transduction pathways implicated in several processes, including apoptosis, proliferation, inflammation and autophagy [63,64,65,66,67].

Two common types of lipid rafts have been described: planar lipid rafts, and invaginated rafts which curve inward, called caveolae. Caveolae are specialized lipid rafts found in very high numbers in adipocytes, which are formed by the polymerization of caveolin1 (CAV1) to generate concave regions on the cell membrane. Caveolae are involved in endocytosis of different proteins and can also play a role in signal transduction, although they may not be essential, since several cell types, such as neurons and lymphocytes, lack caveolin [67,68].

The formation of lipid rafts is not merely confined to the plasma membrane; indeed, similar lipid microdomains, more specifically known as lipid “raft-like microdomains”, are distributed on the membrane of subcellular organelles, which include Golgi apparatus, nuclei, endoplasmic reticulum (ER) and mitochondria [69,70,71,72,73]. Several studies have shown that these organelles differ both quantitatively and qualitatively in their lipid content. Raft-like microdomains are enriched in gangliosides and cholesterol, but with a lower content compared to plasma membranes. In addition, some components of raft-like microdomains are present within ER mitochondria-associated membranes [74]. At these sites, key reactions can be catalyzed, with a significant impact on the regulation of intracellular trafficking, sorting and cell fate [74,75,76].

It is well known that lipid rafts have been associated with several cell functions, including cell death. In fact, it has been suggested that lipid rafts can play a key role in the receptor-mediated apoptosis of T cells. Following CD95/Fas triggering, as well as tumor necrosis factor-family receptors (TNFRs), procaspase-8 oligomerization drives its activation through self-cleavage, activating downstream effector caspases and leading to apoptosis. Thus, activation of Fas results in receptor aggregation and formation of the so-called “death-inducing signaling complex” (DISC) [65,77], containing trimerized Fas, Fas-associated death domain (FADD) and procaspase-8 in lipid rafts, and also causing the recruitment of specific proapoptotic Bcl-2 family proteins to mitochondrial “raft like microdomains”. The importance of lipid rafts in Fas-mediated apoptosis was further supported by the finding that the expression of membrane sphingomyelin, a major component of lipid rafts, enhances Fas-mediated apoptosis through increasing DISC formation, the activation of caspases, the efficient translocation of Fas into lipid rafts, and subsequently Fas clustering [78].

Whether changes in the composition or structure of lipid rafts play a role in autoimmunity is an important question which has started to be addressed in the last few years.

In particular, a role for lipid rafts in T cells activation was reported. In fact, the initial events of T-cell activation involve the movement of T-cell receptor (TCR) into lipid rafts [79,80]. Peripheral blood T cells isolated from SLE patients were found to have more cholesterol and GM1 content in their plasma membrane compared with healthy individuals. This suggests an activated state of T cells in SLE, in which aggregated lipid rafts on the T-cell surface contain TCR as well as other co-stimulatory molecules [81]. Deng et al. reported that disruption of lipid rafts by methyl-beta-cyclodextrin (MβCD) delayed disease progression, whereas aggregation of rafts using cholera toxin accelerated disease progression in mice [82].

In the APS, further studies demonstrated that anti-β2-GPI react with their target antigen, such as β2-GPI, annexin A2 (ANXA2), TLR2 and TLR4 within lipid rafts located in the plasma membrane of monocytes or endothelial cells, thereby producing a proinflammatory, procoagulant phenotype characterized by the release of TNFα and TF, respectively [36,83,84,85]. In Figure 2, the major intracellular signaling pathways triggered by anti-β2-GPI antibodies through lipid rafts and implicated in the procoagulant cell phenotype are depicted. In this regard, the first indication derived from the observation of Sorice et al. [36] is that the anti-β2-GPI target antigen was found within lipid rafts which are specialized portions of the cell plasma membrane implied in signal transduction pathways, as revealed by the sucrose gradient analysis and coimmunoprecipitation experiments.

Interestingly, inappropriate changes in the size and/or structure of lipid rafts could influence their stability and may result in abnormal signaling.

## 4. Signal Transduction Pathway through Lipid Rafts in the Pathophysiology of APS

Signal transduction pathway(s) through lipid rafts play a key role in the proinflammatory and procoagulant events during APS, involving three cell types: endothelial cells, monocytes and platelets. Raschi et al. [35] first demonstrated that anti-β2-GPI antibodies activate the TLR4 transduction signaling pathway in human endothelial cells, as revealed by transiently co-transfecting microvascular endothelial cells (HMEC-1), with dominant-negative constructs of different components of the pathway. Furthermore, the activation of these cells has been demonstrated by the phosphorylation of the myeloid differentiation factor 88 (MyD88), IRAK and NF-κB. Several receptors have been suggested to mediate β2-GPI/endothelial cell binding, including mainly ANXA2 and LRP8 [86]. Engagement of TLR4 activates both MyD88-dependent and MyD88-independent pathways, which may lead to the activation downstream of the MAP kinases. These pioneer studies on the signal transduction pathway triggered by anti-β2-GPI antibodies failed to demonstrate the role of lipid rafts. However, since it is known that plasma membrane TLRs are highly enriched in lipid rafts [87], the involvement of these domains in this pathway was further clarified in human monocytes [36]. In this regard, it was noted that anti-β2-GPI antibody binding on the surface of monocytic cells occurs within lipid rafts [88]. Biochemical analyses showed that the dimeric form of β2-GPI was present within lipid rafts [36], which may be a consequence of an oxidation process [89]. Thus, β2-GPI interacts with lipid rafts only after dimerization, probably as a consequence of conformational changes, suggesting that oxidized β2-GPI is able to trigger signal transduction pathways [89,90] and that β2-GPI dimers mimic in vitro the effects of β2-GPI-anti-β2-GPI antibody complexes [90]. In addition, it was found that ANXA2, the main receptor for β2-GPI [91,92], is highly enriched in lipid raft fractions, wherein it may interact with β2-GPI. It is important to note that, although cell-surface ANXA2 lacks an intracellular tail and is unable to induce intracellular signal transduction by itself, it can act as a binding partner for intracellular surface molecules in lipid rafts [93,94,95]. The involvement of TLR2, as well as TLR4 and the interaction between β2-GPI and ANXA2 within lipid rafts supports the view that TLRs act as “adaptor” proteins with consequent IRAK phosphorylation and activation of NF-κB. These findings indicated that lipid rafts play a key role in the signal transduction pathway induced by anti-β2-GPI antibodies, and that raft-dependent anti-β2-GPI antibody triggering resulted in the release of either TNF-α and TF, which may contribute to the pathogenesis of thrombosis in APS [40,96].

Over the years, a molecular mimicry has been shown between β2-GPI and bacterial antigens, or other various infectious agents [97]. Interestingly, an epitope of *P gingivalis* RNA polymerase s-70 factor shares a complete identity with a peptide of β2-GPI domain I.

Recent evidence showed the presence of autoantibodies specific to this peptide of β2-GPI, which shares a high homology with an extracellular epitope of TLR4, in APS sera. Indeed, TLRs are important components of innate immunity, recognizing specific microbial products and driving the inflammatory response [98]. Antibodies directed to the epitope shared by β2-GPI and TLR4 induced IRAK phosphorylation and NF-kB translocation through the triggering of the TLR4 signaling cascade within lipid rafts, thereby promoting both VCAM expression on endothelial cells and TNF-a release from monocytes responsible for a proinflammatory phenotype [99].

Recent evidence showed that anti-β2-GPI antibodies may also trigger a similar signal transduction pathway in human platelets, which involves IRAK phosphorylation and NF-κB activation followed by TF expression, indicating that platelets may also play a role in the pathogenetic mechanism of APS through lipid rafts [43].

Indeed, TF expression may be induced through the simultaneous activation of NF-κB proteins (via the p38 MAPK pathway) and of the MEK-1/ERK pathway [42]. This may reflect the presence of more than one synergic activating pathway.

## 5. “New” Signaling in the Immunopathogenesis of APS

Although numerous aspects of the molecular mechanisms involved in the immunopathogenesis of APS remain still unknown, recent papers have provided new knowledge of the underlying additional signaling pathways triggered by aPLs, and often these pathways connect the innate immune system and the coagulation cascade [100,101,102]. Following some observations that protease activated receptor 2 (PAR-2) inhibition prevents the expression of TF induced by aPLs [103,104] and that LRP6 has been identified as a novel co-receptor of PAR-2 [105], we described an additional signaling in which anti-β2-GPI antibodies trigger the LRP6 pathway, with consequent phosphorylation of β-catenin [102]. LRP6 belongs to the same family as LRP8, a putative receptor of anti-β2-GPI antibodies [106], but also appears to bind oxide phospholipids involved in pathological conditions such as atherosclerosis and inflammation [107]. LRP6, a type I transmembrane receptor of LDL receptor-related proteins, is a pivotal co-receptor in numerous biological processes. LRP6 binds different Wnt ligands [108], but also several Wnt antagonists, such as Dickkopf 1 (DKK1) [109,110,111]. β-catenin, which represents the key effector of this signaling pathway, is constantly degraded; however, in the presence of Wnt, it translocases to the nucleus, inducing the expression of an array of genes downstream [105,112]. Haack et al. [113] suggested lipid raft involvement in the Wnt/β-catenin pathway. Our findings indicated that anti-β2-GPI antibodies interact with their receptor complex within lipid rafts, leading to the formation of the complex β2-GPI—LRP6—PAR-2 [102]. This hypothesis was strongly supported by coimmunoprecipitation experiments, which revealed that β2-GPI coupled with LRP6 after anti-β2-GPI antibodies treatment. Moreover, the anti-β2-GPI-induced TF expression was prevented by treatment with MβCD, a raft-affecting drug, as well as by treatment with DKK1, a selective inhibitor of LRP6. Thus, this additional signaling pathway seems to be involved in the induction of procoagulant phenotype of endothelial cells.

Thrombosis is the major disease mechanism of APS and vasculopathy, enhanced mainly by severe intimal hyperplasia; it can also play a role in vascular occlusions and pregnancy morbidity [114,115]. Recently, another signaling related to vasculopathy of APS has been described, in which aPLs directly activate the mammalian target of the rapamycin complex (mTORC) pathway in intrarenal vessels of patients with primary and secondary APS nephropathy [100]. Moreover, patients with APS nephropathy who required transplantation and were receiving Sirolimus, an inhibitor of mTORC [116], displayed decreased vascular proliferation. These data have been confirmed by in vitro experiments, where the intensity of mTORC activation in endothelial cells has been shown to be correlated with aPL titers. Furthermore, the exposure of endothelial cells to Sirolimus for 1 h completely inhibited the phosphorylation of S6 ribosomal protein (S6RP) induced by aPLs, but not the phosphorylation of AKT (Ser473). However, cells treated with aPLs and exposed to Sirolimus for 48 h showed reduced phosphorylation of both S6RP and AKT (Ser473). These results are consistent with the activation of both mTORC1 and mTORC2 in endothelial cells, leading to their proliferation and the proliferation of the surrounding vascular smooth muscle cells (VSMC). Moreover, to explain both thrombotic and pregnancy complications of APS, a complex signaling pathway triggered by the interaction between aPLs and the endothelial protein C receptor (EPCR), which is bound to lyso-bis-phosphatidic acid (LBPA), has been recently described [101]. Under normal physiological conditions, EPCR has an important anti-thrombotic and anti-inflammatory effect. EPCR belongs to the MHC class I/CD1 family, and shares with these molecules a feature of lipid exchanging and loading through endosomal recycling [117,118,119]. After the lipid-reactive aPL binding, EPCR internalizes itself. This effect implies the exchange of EPCR-bound phosphatidylcholine, a membrane phospholipid, with LBPA, a late endosomal lipid. aPLs, by binding the EPCR-LBPA complex, may induce the coagulation cascade and activate monocytes and endothelial cells into a pro-coagulant phenotype. The interaction of aPLs with EPCR-LBPA triggers the activation of embryonic trophoblast cells. Moreover, the EPCR-LBPA complex promotes the production of α-interferon by dendritic cells, which eventually sustains a TLR7-dependent expansion of B1a cells and enhances production of aPLs. B1 cells are a rare B lymphocyte subpopulation with unique cell surface antigens that spontaneously secrete IgM. The expansion of B1a cells may be responsible for producing lipid-reactive aPLs in a vicious cycle that enhances the damage mechanisms in APS. The subsequent upregulation of TNF-α and TF plays a key role in the pathogenesis of thrombosis and pregnancy complications. In vivo, blocking EPCR-LBPA signaling in Tlr7−/− mice prevents the development of lipid reactive aPLs and remarkably attenuates thrombosis, fetal loss, and kidney damage [101].

However, several events are likely to play a role in pathogenesis of APS. Recent studies have focused on new signaling pathways responsible for complications of APS that could be useful in understanding the pathogenesis of APS, but also in improving the management of APS patients.

## 6. Potential Therapeutic Strategies through Lipid Rafts

The well-known multifactorial nature of the pathogenic mechanisms of APS opens a broad panorama of potential target molecules. However, at present, therapeutic strategies are aimed at the symptom, and therefore essentially at the treatment of thromboinflammation. In fact, today, the long-term management of patients with APS essentially involves alternative vitamin K antagonists, such as the new oral anticoagulant agents, direct and indirect thrombin inhibitors, hydroxychloroquine, and TF inhibitors [120,121,122,123,124].

The search for new therapies that can guarantee clinical efficacy and at the same time avoid too heavy an impact on the patient’s quality of life is very open. For reasons of efficacy and safety, also linked to the impact of long-term therapy on the patient’s daily life, adherence to treatment and possible food interactions, it is necessary to experiment with new therapeutic strategies. The most recent discoveries on the pathogenetic mechanisms of APS show us new avenues for targeted therapies.

In this concern, as reported in Figure 1, various signaling pathways involved in the procoagulant effect of aPLs act through lipid rafts. Moreover, the signal transduction pathway triggered by anti-β2-GPI antibodies can be strongly influenced by the structure and function of the lipid raft; in fact, modifications in the size and/or structure of lipid rafts would lead to impaired stability and function of these structures and, consequently, could cause signaling abnormalities [125].

Thus, targeting lipid rafts is a rapidly growing therapeutic approach for the treatment of various diseases, from neurodegeneration and neuropathic pain to cancer, infections and atherosclerosis. The approach’s aims to affect lipid rafts can be different; they may be through raising the levels of natural cholesterol acceptors (i.e., HDL) or their mimetic counterparts, or facilitating the efflux of cholesterol with suitable transporters [126,127,128,129,130]. However, the most direct approach consists of depleting the structural lipids responsible for the stability of the raft, such as glycosphingolipids cholesterol or sphingomyelin [125]. In this concern, inhibition of glycosphingolipid biosynthesis in vitro with a clinically approved inhibitor (N-butyldeoxynojirimycin) corrected CD4^+^ T cell signaling and functional defects, and decreased autoantibody production in SLE patients [131].

However, the most common handling approach for affecting lipid rafts includes molecules that are capable of physically sequestering and subtracting cholesterol from the cell membrane [132,133]. In particular, cyclodextrins have been used for some time, both as a food additive (therefore safe for consumption) and as an additive to various pharmacological preparations to increase the stability, solubility and bioavailability of many drugs with anti-inflammatory properties [131]. Cholesterol and sphingolipids can also be depleted from the cell, using inhibitors of their biosynthesis such as statins and Miglustat, respectively [134]. Statins, competitive inhibitors of HMG-CoA reductase, a rate-limiting enzyme of the cholesterol biosynthesis pathway, have long been widely used for the treatment of hypercholesterolemia and represent a concrete possibility of modulating the activity of lipid rafts. Statins are the ideal example of connection with the classical strategy based on membrane cholesterol depletion. They are a well-known class of cholesterol lowering agents and possess several pleiotropic effects (i.e., cholesterol-independent), including the ability to influence the organization of artificial and biological membranes.

Indeed, they are cholesterol-regulating agents that exert anti-inflammatory, immunoregulatory, and anti-thrombotic properties [134,135,136,137]. In vitro and in vivo studies (animal models with APS) demonstrated that statins can inhibit the activation of endothelial cells by aPLs and prevent the overexpression of TF, IL-6 and adhesion molecules [138,139,140,141]. Thus, statins (by inhibition of cholesterol synthesis) or cyclodextrins (by depletion of membrane cholesterol), through affecting cholesterol levels and disrupting lipid rafts, could effectively inhibit the aPLs signaling pathway’s activation (Figure 2).

In conclusion, the progress made in recent years in understanding the biological significance of lipid rafts is indisputable. However, further investigations needed to be conducted, including on the interplay between lipids and membrane proteins in affect membrane organization, in order to provide new therapeutic strategies.

However, targeting the pathogenic pathways of APS development would be crucial, and various studies are underway. A deeper understanding of the immunological mechanisms at the basis of APS is still needed in order to improve the classical anti-thrombotic approaches with “new” immunomodulatory drugs.

## 7. Conclusions

Emerging data pointed out the key role of lipid rafts, specific microdomains on the plasma membrane of endothelial cells, monocytes and platelets, in the signal transduction pathway(s) implied in the pathogenesis of APS. Knowledge of these pathways and clarification of the role of lipid rafts might present new perspectives on the deepening of the immunopathogenesis of the syndrome and on the identification of “new” pharmacological targets that will allow us to move towards personalized medicine.

## Figures and Tables

**Figure 1 jcm-12-00891-f001:**
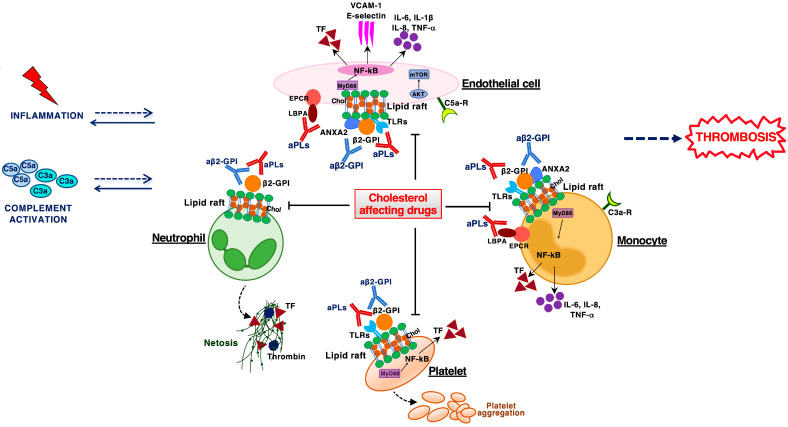
Pathogenetic mechanisms contributing to thrombosis in APS and possible molecular targets. Molecular mechanisms of aPLs, in particular of anti-β2-GPI antibodies, involve the cooperation of endothelial cells, monocytes and platelets with a production of adhesion molecules, proinflammatory cytokines and tissue factor, leading to a proinflammatory and procoagulant state. Anti-β2-GPI antibodies may also activate neutrophils by inducing the formation of NETs important to propagate inflammation and contribute to thrombosis. Inflammation and complement activation play a central role in upregulating cell activation in the pathophysiology of APS. These pathogenetic pathways underlying clinical manifestations of APS can be exploited as targets of immunomodulatory therapeutic strategies to be used in APS patients. Beta2-glycoprotein I (β2-GPI); Annexin 2 (ANXA2); toll-like receptors (TLRs); C3a receptor (C3a-R); C5a receptor (C5a-R); cholesterol (Chol); endothelial protein C receptor (EPCR); lyso-bis-phosphatidic acid (LBPA); myeloid differentiation factor 88 (MyD88); mammalian target of rapamycin complex (mTOR); nuclear factor-kappa B (NF-κB); neutrophil extracellular traps (NETs); vascular cellular adhesion molecule (VCAM-1); intercellular adhesion molecule-1 (ICAM-1); tumor necrosis factor alpha (TNF-α); tissue factor (TF); interleukin (IL).

**Figure 2 jcm-12-00891-f002:**
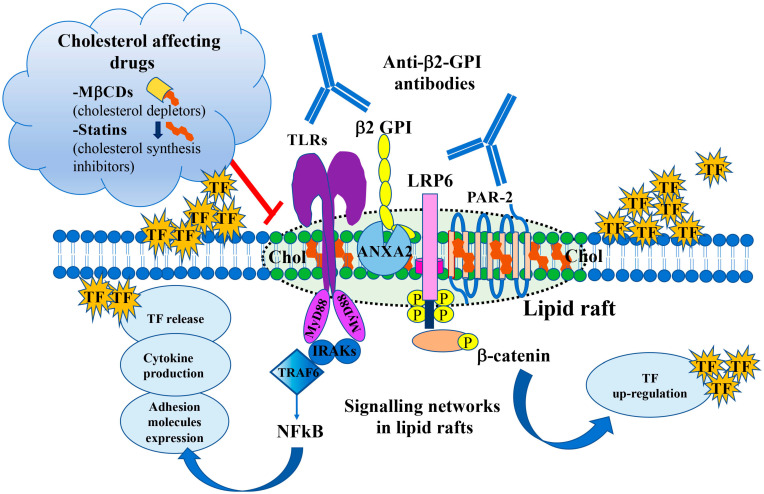
Signaling pathways induced by anti-β2-GPI antibodies. Scheme depicting the major intracellular signaling pathways triggered by anti-β2-GPI antibodies via TLRs and/or LRP6 through lipid rafts (dotted circle), implicated in the procoagulant endothelial cell phenotype characterized by the expression of TF. Lipid raft-targeted drugs act on cholesterol either as cholesterol depletors (MβCDs) or cholesterol synthesis inhibitors (statins). Annexin A2 (ANXA2); anti-β2-glycoprotein I (anti-β2-GPI antibodies); beta-catenin (β-catenin); cholesterol (Chol); interleukin-1 receptor–associated kinases (IRAKs); LDL receptor related protein 6 (LRP6); methyl-β-cyclodextrin (MβCD); myeloid differentiation factor 88 (MyD88); nuclear factor-kappa B (NF-κB); protease activated receptor-2 (PAR-2); tissue factor (TF); toll-like receptors (TLRs); TNF receptor associated factor 6 (TRAF6).

## Data Availability

Data sharing not applicable.

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
