# Peer review of "Advances in the Pathophysiology of Thrombosis in Antiphospholipid Syndrome: Molecular Mechanisms and Signaling through Lipid Rafts"

_jcm, 2023, doi:10.3390/jcm12030891_

Round 1
Reviewer 1 Report
Dear Authors,
I've read with great interest your review article entitled ''Advances in the Pathophysiology of Thrombosis in Antiphospholipid Syndrome: molecular mechanisms and signaling through lipid rafts. In this manuscript, you comprehensively summarize the pathophysiology of APS and describe the potential roles of lipid rafts in this condition.
This review is a good overview of the topic and adds value to the scientific audience.
I have a few minor comments:
1. Line 272: '' In the last few years'' is a misleading statement, as the reference [98] investigating peptide from domain-I of B2GPI and some infectious antigens dates in 2004... I would suggest replacing this ref. with some newer evidence or mitigating the statement.
2. The sentence line 276-277: ''Recent evidence showed in APS sera the presence of autoantibodies specific to this peptide, that shares a high homology with an extracellular epitope of TLR4.'' is confusing. In fact, the whole paragraph 276-282 is hard to understand. What did the authors mean by ''These antibodies''? Are these anti-TLR4 antibodies? Please clarify this paragraph.
3. Chapter 5: Potential therapeutic strategies through lipid rafts. The first few paragraphs are repeating the previous content and explanations therefore they could be omitted or shortened.
Author Response
I've read with great interest your review article entitled ''Advances in the Pathophysiology of Thrombosis in Antiphospholipid Syndrome: molecular mechanisms and signaling through lipid rafts. In this manuscript, you comprehensively summarize the pathophysiology of APS and describe the potential roles of lipid rafts in this condition.
This review is a good overview of the topic and adds value to the scientific audience.
We thank the reviewer for his/her positive comment.
Minor comments:
- Line 272: '' In the last few years'' is a misleading statement, as the reference [98] investigating peptide from domain-I of B2GPI and some infectious antigens dates in 2004... I would suggest replacing this ref. with some newer evidence or mitigating the statement.
We modified the text accordingly, reporting “Over the years”.
- The sentence line 276-277: ''Recent evidence showed in APS sera the presence of autoantibodies specific to this peptide, that shares a high homology with an extracellular epitope of TLR4.'' is confusing. In fact, the whole paragraph 276-282 is hard to understand. What did the authors mean by ''These antibodies''? Are these anti-TLR4 antibodies? Please clarify this paragraph.
We clarified the text, indicating that antibodies are directed to the epitope shared by β2-GPI and TLR4.
- Chapter 5: Potential therapeutic strategies through lipid rafts. The first few paragraphs are repeating the previous content and explanations therefore they could be omitted or shortened.
We agree with the comment of the reviewer and omitted the first few paragraphs in the revised manuscript.
Reviewer 2 Report
In this paper, the authors performed a review of the advances in the pathophysiology of thrombosis and antiphospholipid syndrome. They focused on molecular mechanisms and signaling pathways through lipid rafts.
The paper written by a recognized team in the field is clear and very well-written. The bibliography is well-documented.
I have some minor suggestions:
- The authors should describe more precisely the different receptors according to the cell type involved in the signaling intracellular pathways. The involvement of different receptors could explain the different clinical manifestations (paragraph 1: mechanism in the pathophysiology of APS).
- The authors should describe more precisely how the involvement of lipid raft was dicovered in APS (paragraph 2: lipid rafts in the immune signaling).
Author Response
In this paper, the authors performed a review of the advances in the pathophysiology of thrombosis and antiphospholipid syndrome. They focused on molecular mechanisms and signaling pathways through lipid rafts.
The paper written by a recognized team in the field is clear and very well-written. The bibliography is well-documented.
We thank the reviewer for his/her positive comment.
Minor suggestions:
- The authors should describe more precisely the different receptors according to the cell type involved in the signaling intracellular pathways. The involvement of different receptors could explain the different clinical manifestations (paragraph 1: mechanism in the pathophysiology of APS).
We added in the paragraph 1 the different receptors involved in the signaling intracellular pathways triggered by the antibodies.
- The authors should describe more precisely how the involvement of lipid raft was dicovered in APS (paragraph 2: lipid rafts in the immune signaling).
We better described in the paragraph 2 how the involvement of lipid raft was discovered in APS.
Reviewer 3 Report
In this current review, the authors provide some updates of pathophysiology, and insights focused on lipid rafts in APS. The contents include the mechanism in the pathophysiology of APS and signal transduction pathway through lipid rafts in the pathophysiology of APS and potential therapeutic strategies through lipid rafts.
However, this review included redundant parts unrelated to lipid rafts, leading to complicated contents. In addition, the explanations of figures are insufficient.
1. The authors should compact “1. Mechanisms in the pathophysiology of APS”.
2. Figure 1 included both “pathogenetic mechanisms” and “possible molecular targets”. However, “possible molecular targets” described away from “pathogenetic mechanisms” in the manuscript. In addition, there is no advanced information in figure 1. The authors should amend Figure1 or structure of the manuscript.
3. In Figure2, the explanations of figures are insufficient. What is green circle? Why are four antibodies on this figure? Where is mTORC or EPCR-LBPA? Why do lipid rafts targeted drugs inhibit signaling networks in lipid rafts?
4. “5. Potential therapeutic strategies through lipid rafts” included redundant contents unrelated to lipid rafts. The authors should remove the redundant contents and focus on lipid rafts.
Author Response
In this current review, the authors provide some updates of pathophysiology, and insights focused on lipid rafts in APS. The contents include the mechanism in the pathophysiology of APS and signal transduction pathway through lipid rafts in the pathophysiology of APS and potential therapeutic strategies through lipid rafts.
However, this review included redundant parts unrelated to lipid rafts, leading to complicated contents. In addition, the explanations of figures are insufficient.
We thank the reviewer for these comments. In the paragraph 5 of the revised manuscript, we omitted the redundant parts unrelated to lipid rafts. Furthermore, we modified both Figures focusing on the role of lipid rafts and improved the explanation of Figures.
- The authors should compact “1. Mechanisms in the pathophysiology of APS”.
We compacted paragraph 1.
- Figure 1 included both “pathogenetic mechanisms” and “possible molecular targets”. However, “possible molecular targets” described away from “pathogenetic mechanisms” in the manuscript. In addition, there is no advanced information in figure 1. The authors should amend Figure1 or structure of the manuscript.
We modified Figure 1 focusing on the pathogenic mechanisms. We improved the information, adding new key molecules, such as mTOR and EPCR-LBPA.
- In Figure2, the explanations of figures are insufficient. What is green circle? Why are four antibodies on this figure? Where is mTORC or EPCR-LBPA? Why do lipid rafts targeted drugs inhibit signaling networks in lipid rafts?
We modified and clarified Figure 2. The green circle has been modified (now dotted circle). We indicated that it represents the lipid raft. We now show only two molecules of antibodies activating the signaling pathways to avoid misunderstanding. We added mTOR and EPCR-LBPA in Figure 1, because Figure 2 is specifically focused on signaling networks in lipid rafts. We clarified that lipid rafts targeted drugs act on cholesterol either as cholesterol depletors (MbCDs) or cholesterol synthesis inhibitors (statins).
- “5. Potential therapeutic strategies through lipid rafts” included redundant contents unrelated to lipid rafts. The authors should remove the redundant contents and focus on lipid rafts.
We thank the reviewer for this comment. In the paragraph 5 of the revised manuscript we removed the redundant contents and focused on lipid rafts.
Round 2
Reviewer 3 Report
This manuscript does not provide sufficient responses to the reviewer's comments.
Author Response
This manuscript does not provide sufficient responses to the reviewer's comments.
We further improved the manuscript according to the previous comments of the reviewer.
New reply to the reviewer’s comments:
In this current review, the authors provide some updates of pathophysiology, and insights focused on lipid rafts in APS. The contents include the mechanism in the pathophysiology of APS and signal transduction pathway through lipid rafts in the pathophysiology of APS and potential therapeutic strategies through lipid rafts.
However, this review included redundant parts unrelated to lipid rafts, leading to complicated contents. In addition, the explanations of figures are insufficient.
We further revised the paragraph on potential therapeutic strategies through lipid rafts, omitting redundant parts unrelated to lipid rafts. Furthermore, we further improved the explanation of Figures, including two sentences in the text (lanes 91-94 for Figure 1 and 227-229 for Figure 2) and one in the legend of Figure 2 (lanes 240-241).
- The authors should compact “1. Mechanisms in the pathophysiology of APS”.
We reorganized the old Paragraph 1 in two parts: Introduction with general information on APS and a second one focused on the mechanisms in the pathophysiology of APS.
Figure 1 included both “pathogenetic mechanisms” and “possible molecular targets”. However, “possible molecular targets” described away from “pathogenetic mechanisms” in the manuscript. In addition, there is no advanced information in figure 1. The authors should amend Figure1 or structure of the manuscript.
Figure 1 had been modified and improved in the Revised manuscript, focusing only on the pathogenic mechanisms and deleting all the potential therapeutic approaches unrelated to lipid rafts.
In Figure2, the explanations of figures are insufficient. As reported above, we further improved the explanation of the Figure, including a new sentence in the text (lanes 227-229) and in the legend (lanes 240-241).
What is green circle?
The green circle indicates the lipid raft. We had modified the graphical presentation (dotted circle), clarifying that it identifies this specific area of plasma membrane.
Why are four antibodies on this figure?
We now show only two molecules of antibodies which schematically indicate the activation of the two main signaling pathways triggered by autoantibodies within lipid rafts.
Where is mTORC or EPCR-LBPA?
We had added mTOR and EPCR-LBPA in Figure 1, because Figure 2 is specifically focused on signaling networks in lipid rafts.
Why do lipid rafts targeted drugs inhibit signaling networks in lipid rafts?
We clarified that lipid rafts targeted drugs act on cholesterol either as cholesterol depletors (MbCDs) or cholesterol synthesis inhibitors (statins). We had added this clarification in the cloud in Figure 2 and now we added a new sentence in the legend (lanes 240-241).
- “5. Potential therapeutic strategies through lipid rafts” included redundant contents unrelated to lipid rafts. The authors should remove the redundant contents and focus on lipid rafts.
We thank the reviewer for this comment. We fully further revised the old paragraph 5. In particular, we reorganized it, focusing on potential therapeutic strategies through lipid rafts, further removed the redundant contents and improved the description of raft affecting drugs. In this concern, we better clarified the mechanism of action of statins and added a new clinically approved inhibitor of glycosphingolipid biosynthesis (see new Ref 132).